# Situation awareness of emergency response centre personnel during chemical incidents: an interview study in a Swedish context

Sofia Karlsson [ORCID],[1] Lina Gyllencreutz [ORCID] [1,2]

¹Umeå University, Department of Surgical and Perioperative Sciences, Surgery, Umeå, Sweden
²Umeå University, Department of Nursing, Umeå, Sweden

**Correspondence to**
Dr Lina Gyllencreutz;
lina.gyllencreutz@umu.se

## ABSTRACT

**Objectives** If a chemical incident occurs, the emergency response centre (ERC) personnel are the first that are notified. They need to quickly attain situation awareness, based on the information from the caller, in order to dispatch the correct emergency units. The aim of this study is to examine the situation awareness of the personnel working at ERCs—how they perceive, comprehend, project and act during chemical incidents.

**Methods** Semi-structured individual interviews with 12 participants from the Swedish ERCs were performed. The interviews were analysed with qualitative content analysis.

**Results** Three categories of responses were identified. Responses focused on the complexity of identifying chemical incidents, the importance of ensuring the safety of citizens and personnel of emergency organisations and the situation-based dispatch of organisations.

**Conclusions** The correct identification of the chemical incident and the involved chemical by the ERC personnel are necessary in order to notify, inform and dispatch the correct units, as well as to ensure the safety of citizens and emergency personnel. More research is needed about the dichotomies of the ERC personnel needing as much information as possible for everyone's safety versus their responsibility for the safety of the caller as well as between using emergency dispatch index interview guides and trusting their gut feeling.

## INTRODUCTION

Worldwide, large quantities of chemical substances are produced, transported and used, making incidents probable.[1 2] Incidents can result from human error or poor maintenance of equipment used for manufacturing and storage.[3] Incidents can include fires, explosions, leaks and spills and structural collapse.[3] Chemical incidents involving the uncontrolled release of toxic substances can potentially lead to injury or death of exposed persons, as well as harm the environment.[1] A previous chemical incident that killed and injured many people was the Beirut port explosion in Lebanon on 4 August 2020.[4] However, most chemical incidents are small or medium-sized, and generally involve few

killed or injured persons,[1 3] exemplified by a fire in a Swedish battery factory and an ammonia spill that injured about 10 persons, respectively.[5 6]

⇒ The combination of theory and experiences from emergency response centre (ERC) personnel on situation awareness in the event of a chemical incident strengthens the confirmability of this study.
⇒ The geographical diversity of the participants in this study, as well as their differing professional roles strengthen the credibility.
⇒ The interview guide is transparent and therefore increases dependability, including consistency and repeatability of the study.
⇒ A limitation of this study's transferability is that, due to a diverse system, the Swedish ERC personnel have different roles, responsibilities and education, depending on what ERC they operate at.

Internationally as well as in Sweden, when an incident occurs, the first point of contact with the emergency organisations is often the emergency response centres (ERC). The ERC operator is tasked with gaining as much relevant information from a caller as possible, and with determining which response is required.[7] Dispatchers are connected to the call to inform and dispatch the correct emergency organisation units.[7–9] The ERC personnel swiftly manage a large quantity of information of varying quality from different sources, such as the general public or on-site emergency personnel, in order to make decisions in a dynamically changing situation.[10] Thus, in order to do this, ERC personnel use their situation awareness, which means they perceive elements in the environment, comprehend the current situation and project its future status.[11] The ERC personnel form a mental picture of the situation, where they use their experience and intuition in order to decide how to properly deal with a

call.[12] Because every call is unique, ERC personnel must be sensitive listeners, and should be able to reformulate their questions depending on the situation, balancing their use of the interview guide and their common sense and intuition.[13 14] However, an initial lack of information about the incident can result in injured first responders, as in the case of the release of anhydrous ammonia from a farm tractor towing a 2-ton ammonia tank experiencing mechanical failure in Illinois, USA.[15] The ERC personnel initially reported the incident as a car fire, which led to several first responders being unaware of the danger. A number of responders were thus exposed to the gas, and some later experienced symptoms of illness.[15] Thus, the ERC personnel's situation awareness, related to their ability to identify the incident and predict medical outcomes for those affected, is highly important in chemical incidents.

The literature states that further study is needed on the challenges of situation awareness and behaviours used by the ERC personnel to overcome those challenges.[16] Further, few studies focus solely on the ERC personnel's experiences of chemical incidents. Because of the prevailing danger during chemical incidents for the affected persons and responders alike, ERC personnel need to be able to manage the incoming information. This is especially true during major chemical incidents, when many persons may potentially be injured. Thus, the aim of this study is to examine the situation awareness of the personnel working at ERCs—how they perceive, comprehend, project and act during chemical incidents.

## METHOD
### Setting
In Sweden, ERC is an important part of the national emergency preparedness responding to approximate 3 000 000 calls a year. Depending on which ERC the personnel work at, they have different roles, responsibilities and education. The personnel at the ERC work either as an ERC operator, rescue service dispatchers or emergency medical service dispatcher. Some personnel are authorised to work in several positions, for example, variously working both as an ERC operator and a rescue service dispatcher.

The ERC operators are responsible to answer the national emergency number 112 and interview the callers based on index interview guide, that is, a criteria-based dispatch protocol structured in subindices. The dispatchers also listen to the call and can guide the operators to ask additional questions based on their specific expertise. The dispatchers should then dispatch and coordinate the emergency medical service and rescue service units. The ERC operators are also collaborating with other organisations, such as the police.[8 17] In some Swedish regions, registered nurses perform the medical interview instead of an ERC operator.[8 18] The registered nurses use more unrestricted guidelines with prompts based on caller descriptions of signs and symptoms, to provide direction and assistance in defining appropriate levels of care. In this study we have interviewed both operators, registered nurses and dispatchers.

### Participants
Information about the study was provided in January 2021 to five managers of the ERCs located in various parts of Sweden. The managers notified their personnel of the study, and interested personnel with experience of chemical incidents were invited to participate. No time limit on when the incident occurred was considered necessary as the participants themselves chose the incident which they had experience of. In total, 12 respondents participated.

Nine of the participants were women and three were men. Their ages ranged from 32 to 63 years (with a mean age of 43), and their work experience at the ERCs varied from 2 to 32 years (with a mean length of 8 years). Eleven participants worked as ERC operators. Of them five also worked as emergency medical service dispatchers whereof four were registered nurses. Additionally, five operators also worked as rescue service dispatchers. The last operator had exclusively that role. The 12th participant exclusively worked as rescue service dispatcher.

### Data collection
The semi-structured interview guide (see online supplemental file 1) was developed by the authors. The guide is inspired by the theory of situation awareness, applying the concepts of perception of elements in the current situation, comprehension of the current situation, projection of future status and decision-making[11] to the ERC personnel's work during chemical incidents. In order to qualitatively capture the participants' own experiences in dealing with chemical incidents, the opening question was 'Can you describe a situation where you received an emergency call regarding an incident, immediately or later classified as a chemical incident, where there was a risk of people being injured or dying'. The interview questions then referred back to the described incident and situation awareness regarding that incident. The interview guide also contains questions that related to situation awareness and the participant's general chemical preparedness. A test interview was performed by the first author that enabled the development of the interview guide. All interviews were performed via Zoom or Teams. The interviews were performed between February and September 2021 by the first author and a research assistant. The interviews ranged from 34 min to 72 min, with a mean of 63 min. All interviews were recorded and transcribed verbatim by the first author and the research assistants.

### Data analysis
The individual interviews were analysed with qualitative content analysis[19–21] using Microsoft Excel (2019) as management support. The transcripts were first read in order to gain an overall understanding of the content. Meaning units relating to the aim were extracted from

the text and used to create codes. The codes were sorted into categories and subcategories. Relevant citations were used to illustrate the codes and categories. The first author performed the coding and categorisation. The categorisation was then discussed with the second author and iteratively improved. A summary in the beginning of the results has also been added to describe the general situation awareness of the ERC operators. This was done since much of what the ERC personnel say they do in all emergency calls will also influence their work during chemical incidents.

### Ethical considerations

This study was performed in accordance with the Helsinki Declaration and the Swedish Code of Statutes.[22 23] Participants were informed of the study objectives, both verbally and in writing, and were guaranteed confidentiality and the right to withdraw from the study at any time. The participants gave their written informed consent to participate in the study.

### Patient and public involvement

Patients and/or the public were not involved in the design, or conduct, or reporting, or dissemination plans of this research.

### RESULTS

Before going through the results relating to chemical incidents in more detail, a brief summary of the general situation awareness of the ERC personnel influencing their work during chemical incidents will be presented. The general idea of what situation awareness is for the ERC personnel means in this result that they usually quite quickly ascertain whether a given situation deviates from normal patterns. Operators described that they can do this because of their experience of hearing details, such as what is happening in the background of the call, or nuances in the phrasing and pitch of the callers. They described themselves as detectives who trust their gut feelings and judged that their gut feeling was correct in most of the cases. If something felt wrong, personnel followed up with probing questions. The ERC operators also asserted the situation awareness that they had to be able to adapt their style of interviewing depending on the caller—for example, by being empathetic or authoritative, but always being calm and professional.

This study resulted in three categories of responses (table 1), centring on (1) the complexity of identifying chemical incidents, (2) the importance of ensuring the safety of citizens and personnel of emergency organisations and (3) the situation-based dispatch of organisations.

### The complexity of identifying chemical incidents

The ERC personnel assert that it is often difficult to gain situation awareness about whether a chemical incident is actually in progress, what kind of chemicals are involved, how dangerous the situation is and its potential consequences.

#### Perceiving the chemical incident and its severity

Chemical incidents were considered as rare but potentially severe events, varying from singular chemically injured persons to major chemical incidents. The ERC personnel considered major chemical incidents as likely, and recognised that they have to be prepared for them. They considered themselves reasonably well prepared, and asserted that chemical incidents were not more difficult to manage than other incidents. However, participants stated that they became more focused and had to think outside the box when such incidents occurred, especially if they are unsure about the chemical reactions taking place, or in particularly difficult situations such as fires or incidents at sea.

Because of the rarity of chemical incidents, the ERC personnel stated that there was a risk that they would not perceive that a given incident indeed was chemical in nature. Thus, in order to not dismiss a dangerous situation, participants mentioned that they spend more time on incidents they perceived as vague than on obvious ones. Participants mentioned that they have certain trigger words for chemical incidents, which were emission, leakage, traffic incident, strong strange smell, if people are feeling sick without a clear cause, smoke or gas leaks, dripping sounds, that something looks strange, fires in industries or cars or whether a heavy goods vehicle is involved in the incident.

Chemical incidents were also considered as vague incidents because the chemicals can be invisible, which makes it difficult to assess the volume of the release. Participants also mentioned the difficulties of understanding the chemical incident's potential to escalate and become more dangerous in a short time. For example,

**Table 1** Categories and subcategories of interview responses

| Category | Subcategory |
| --- | --- |
| The complexity of identifying chemical incidents | Perceiving the chemical incident and its severity |
| | Identifying chemical substances based on the situation |
| The importance of ensuring the safety of citizens and personnel of emergency organisations | Giving advice to the caller about safety |
| | Considering the safety of the emergency organisations' personnel |
| The situation-based dispatch of organisations | Balancing the need for units with their availability |
| | Considering which other organisations to connect to the emergency call |

incidents can escalate if a fire starts or spreads, if there is a combination of cargo, if the chemical substance comes into contact with water, or if the incident has occurred on a heavily trafficked road.

Another kind of difficult situation entailed having to identify symptoms (such as affected breathing, cough, stinging eyes, nausea, headache and dizziness) as being caused by a chemical incident, because these are such common symptoms. Other ERC personnel considered that chemical incidents could be obvious—for example, incidents with a heavy goods vehicle transporting dangerous chemicals. The ERC personnel considered that it depended on the situation if they clearly comprehended that it was a chemical incident or not—some callers may be very verbal and factual regarding relevant information, while others may be less communicative when anxious:

> They called and said that a person had inhaled chlorine gas at an ice-hockey stadium, which is not strange, it can happen. It never emerged in the call that it was still leaking. The callers never mentioned this, they were mostly shocked. Then the ERC operator had to ask "is it still leaking?". It took about 20-30 seconds to ask that question, because it didn't emerge spontaneously. (Rescue service dispatcher)

### Identifying chemical substances based on the situation

ERC personnel asserted that it was important to quickly identify the chemical substance or substances in play, as well as how dangerous it or they were, but also felt that this was not always easy. One ERC operator mentioned that the caller is their eyes on site. In order to assess the situation and find out which chemical is involved, there are a lot of questions the ERC operators could ask—for example, what does the smoke/substance look like, how does it smell or if observers can see any orange warning signs at heavy goods vehicles. The ERC operators acknowledged that they could be ignorant about chemicals and their effects, and could be in need of more information from the rescue service, the specialist emergency medical service for chemical incidents or the Swedish Poisons Information Centre. The rescue service dispatchers mentioned that they were knowledgeable about chemicals and could also search for more information in the chemical substance database.

> If we receive a chemical formula that we don't recognise, we search the chemical substance database in order to be able to notify the rescue service when they are on their way. We get to know that the chemical substance can react with oxygen and can be ignited at 27°C and it is 22°C outside. This means lives are at risk for those who are at the site of the incident. (ERC operator)

The ERC personnel felt that the general public had minimal knowledge about chemicals and thus preferred if professional personnel (eg, heavy goods vehicle drivers and industry workers) made the emergency call. The professionals are trained in what to say—they know which chemical substances they handle, and are knowledgeable about which risks a given substance entails. In certain cases, for example, if a driver is injured, the ERC personnel could call the transportation business to ask what the heavy goods vehicle is transporting. However, during major incidents, for example, major industry incidents, the ERC operators stated that it could be really difficult to gain information from the professional personnel. They mentioned that their first operational picture could change really quickly while the call progressed and they received more information.

### The importance of ensuring the safety of citizens and personnel of emergency organisations

The ERC personnel stated that they have to attain situation awareness about the chemical incident in order to ensure the safety of both the citizens at the incident site and the emergency organisation personnel that will arrive at the incident site.

### Giving advice to the caller about safety

The ERC operators considered it difficult to give advice to callers during chemical incidents, but acknowledged that they had enough experience and common sense to give sound advice. Most of the time, operator advice was generic, but operators could also give advice from a chemical substance database or based on recommendations from the rescue service. Examples of advice were to move further away from the incident and be mindful of the direction of the wind, to evacuate buildings and wait for the rescue service to rescue trapped persons.

While the ERC operators volunteered that they wanted to receive as much information as possible, they still saw their prime responsibility as ensuring the safety of the caller. Thus, they have to be careful when asking callers to investigate the scene—for example, in case of missed warning signs. If there are others injured at the scene of the incident, they are a secondary priority if there is a risk of even more casualties. Operators usually had to be very straightforward with the callers in telling them to think of their own safety, leave injured persons behind and advise them against running into dangerous situations, as well as in advising callers to keep distance while waiting for experts with appropriate equipment. Sometimes the caller wants to help but does not see the seriousness of the situation, and this might lead to an even worse situation, despite their good intentions.

> When it comes to animals which may burn to death inside a barn or when there are still people left in a burning building, those scenarios are difficult, because people want to run in, most of them anyway. We want them to keep as far away as possible in order for them not to hurt themselves. (ERC operator)

### Considering the safety of the emergency organisations' personnel

The ERC personnel considered it important that they correctly inform the emergency organisations of the chemical incident. Despite not always knowing the chemical substance involved, participants verify that the emergency organisations perceive that it is a dangerous situation. Thus, ERC personnel would rather give a bit too much information than too little, so that the emergency organisations can make their own assessments of the situation. If ERC personnel have knowledge of the chemical substance involved, they can give relevant safety advice, for example, on safety distances. When they do not know the chemical substance involved, the dispatchers considered it to be difficult to indicate to the emergency organisations that a chemical incident actually is in progress. If they do not know the chemical substance(s) involved, or have very little information about the incident, ERC personnel report that they will work as if the worst possible situation prevailed, and then (if called for) de-escalate the situation, based on information from the emergency organisations at the incident site.

> Interviewer: If you had received information that it was hydrochloric acid, what decisions would you then have made?
>
> Rescue service dispatcher: I probably would not really have made any other decisions. However, I could have planned the rescue operation and what resources were necessary. I probably would not have needed as many resources, they could have extinguished the fire earlier at a closer range.

### The situation-based dispatch of organisations

During chemical incidents, the ERC personnel need situation awareness in order to interpret the situation, as well as to inform and dispatch the required organisations. Participants believed that more resources are needed during chemical incidents than during other incidents, but also emphasise that limits on resources must also be taken into consideration.

### Balancing the need for units with their availability

The dispatchers decide which emergency organisation units should be dispatched to an incident based on the first incoming information from the callers, pre-made plans or directives and the competences of the dispatched units (eg, the specialist emergency medical service for chemical incidents). This means they have to dare to make quick decisions about how many units they should dispatch to the incident.

> The rescue service would rather dispatch 10 times too many units than miss dispatching units when we could have saved someone. We are drilled that if we happen to dispatch two rescue service units too many, we can just withdraw them if they are not needed. (Rescue service dispatcher)

The dispatchers' decisions about the number of dispatched units differ depending on the situation—for example, the risk for further injuries, quantity of chemical substances involved and type of environment. Participants assert that chemical incidents demand more units than other incidents—something which often requires that extra emergency medical service units be sent, because there is a risk of injury to rescue service personnel. If a major incident is in progress, dispatchers will send all the available units in the region. However, the emergency medical service dispatchers considered they had to prioritise where they send their units, because they also have a responsibility to care for all other patients with severe injuries and illnesses. For example, in a city they may be able to send all available units to a chemical incident, but not in the countryside because of long distances.

The dispatchers report working with the resources they have available—for example, moving resources in the region to strategic places in order to cover the entire region. They can also collaborate with other regions in order to increase the available resources. The dispatchers will also have a dialogue with the emergency organisations at the incident site if more resources are needed and they can give suggestions of additional available resources. It is the incident commanders who decide which resources are needed based on the situation at the incident site, in relation to the available resources.

### Considering which other organisations to connect to the emergency call

The ERC operators report that they quickly have to consider, contact and inform several organisations and chemical experts, apart from the rescue service and emergency medical dispatchers. For example, they report that there are specialised rescue services which are contacted if an incident happens within their area of responsibility, such as the sea, mountain, industry or airport rescue service. The ERC operators also described that they contact, ask for more information and inform the police response centre operators that dispatch police units to the chemical incident.

> When I understand that it's an incident with several [people] exposed to the same substance, I contact the rescue service and the police. I ask them if they know anything and give them the information we have, that there are several persons that have been exposed and we suspect that it's this substance but we have no idea, but there are people that have been in a specific place and they start to show symptoms and something isn't right. (ERC operator)

Other relevant representatives the ERC personnel mentioned that they can consider contacting, including officials on call, the salvage officer and representatives from the Swedish Transport Administration. The officials on call are contacted for resource allocation and preparation of the hospitals in the region. The salvage officer works together with the rescue service to protect

the environment—for example, by taking care of spilled chemicals and by documenting the incident. Representatives from the Swedish Transport Administration are contacted if the ERC personnel consider it important to block a road or reroute traffic. If there is a major chemical incident, they also contact the Swedish Civil Contingencies Agency, who inform the citizens of the chemical incident through an Important Public Announcement. This could, for example, happen if there is a risk to the health and safety of the citizens due to toxic smoke, or if an area is blocked off due to the risk of explosion.

## DISCUSSION

This study aimed at examining the situation awareness of the personnel working at ERCs—how they perceive, comprehend, project and act during chemical incidents. The three categories of the results will be discussed further below.

Starting with the category of the complexity of identifying chemical incidents, the result shows that it could take time to ascertain whether a situation was dangerous or had the potential to escalate. Prior research has also confirmed that ERC personnel can receive inadequate information or have difficulties in grasping the situation.[24] During chemical incidents this can be especially true if the incident type is not covered by the emergency dispatch index interview guide, or if the information about the incident is fragmented and several callers have to provide information for a full understanding of the situation, as reported in the study by Westman et al.[25] In situations having a clear emergency dispatch index, a swift response can be expected, thus dispatch indexes covering chemical incidents could be developed. As this result shows the ERC personnel's gut feeling is central to picking up on cues that there has been a chemical incident. This is consistent with Leonardsen et al,[12] who state that ERC operators refer to a mental picture where they use their experience and intuition in order to decide how to properly deal with a call. Thus, the results agree with what Ek and Svedlund state,[13] that the ERC operators have to balance between strictly using the interview guide and using common sense and intuition.

Prior literature also indicates that it is especially important to be vigilant during motor vehicle collisions, heavy goods vehicle incidents or structural fires, because, for example, the hazmat placard could have been damaged in the heavy goods vehicle crash.[3] The result in this study shows that hazard plaques and environmental clues could be of great help, but sometimes the owner of the heavy goods vehicles involved needed to be contacted if it was uncertain which chemical substance(s) was involved. This was sometimes complicated by the fact that the driver was unconscious and unable to answer any questions.[3] A suggestion to further improve the knowledge of the ERC personnel on chemical incidents and implement lessons learnt could be to develop emergency

organisation use of chemical incident databases, similarly to what is suggested in the chemical industry literature.[26]

Another result of this study shows that it is easier to receive the emergency call from staff working at an affected site as the information from them is more accurate and clear and thus more useful for the operators than from other callers. This is somewhat similar to what Holmström et al[27] described, that it is more difficult to understand information and assess the severity of the symptoms from other callers, for example, relatives or strangers at an incident site, who do not have any background information about the sick or injured persons. Thus, improving the public knowledge and ability to act during chemical incidents by performing pre-incident information campaigns about chemical incidents[28] could possibly lead to ERC personnel gaining more useful information from the public.

The category 'The importance of ensuring the safety of citizens and personnel of emergency organizations' emphasises the importance of ERC personnel's situation awareness for their ability to give correct information and sound advice, because the chemical incident site may still be dangerous. Prior literature also states that it is important for ERC personnel to be able to assess risks, reassure and give the caller appropriate instructions and safety advice, as well as to gather and share information with the emergency organisations in order to prepare them—both tasks, when executed, will decrease the risks to all involved.[16 24 25 29] This is important because it is relatively common that the personnel of the emergency organisations are injured during chemical incidents.[30] Thus, the ERC personnel of this study stated that it was important to inform the emergency organisations about the situation at the incident site, including information about the chemical substance if they knew it.

Further, prior literature has also pointed out that if the chemical substance in play is known, so is the route of exposure and the likelihood for secondary contamination of the emergency organisation personnel, as well as the most effective method of protection and decontamination.[3] Further, stressing the importance of correct information from the ERC personnel,[31] states that there is a risk that the emergency organisations are misguided, because they form their impressions of the incident based on pre-arrival information rather than post-arrival information. This is also discussed in another study, which states that incomplete information from the ERC personnel could lead the emergency medical service units to misinterpret the situation and lead to wrong actions, which in turn leads to delayed care of those affected.[32] Thus, the importance of the ERC personnel to ensure the safety of both citizens and emergency personnel cannot be stressed too much.

Roud et al has found that so-called reflection seminars can be helpful in focusing on unsolved problems in structures, behaviours and working methods, as well as in confirming existing knowledge and procedures.[33] Such seminars may also build trust and collaboration between

emergency organisations, and complement learning obtained from full-scale exercises.[33] Thus, in order to decrease the risk of injuries to emergency personnel and citizens, it could potentially be of value to create forums for all responding organisations including the ERC personnel, where learning from each other about safety during potential chemical incidents could be discussed. Lessons learnt could then be further inculcated and put into practice during full-scale exercises.

Finally, the importance of the situation-based dispatch of organisations was in this study confirmed by ERC personnel reports that they usually worked with a given understanding of which organisations and units needed to be informed and dispatched to the chemical incident. However, although the ERC personnel in this study stated that chemical incidents demanded a lot of units, they also had to take into consideration how many units they actually had available and prioritise between patients. This is consistent with what other authors state, that ERC personnel have to consider that there are limited resources and that they had to prioritise the calls in relation to other waiting calls.[12 13] Thus, also prior literature states that ERC personnel have to be aware of what is happening in the area regarding available resources and other ongoing incidents in order to send sufficient resources.[7]

Another issue relating to the issues of limited resources is that chemical incidents can be perceived as vague, as shown in this study's results. One prior study has also found that sending resources to vague but urgent alarms could be problematic if the situation responded to was found to be non-urgent on arrival.[29] Thus, ERC personnel have to balance the need of resources with the availability of resources. Reflection seminars could provide a forum where ERC personnel can discuss resource management during chemical incidents with representatives from the emergency organisations.

### Strengths and limitations

One strength of the study is that the participants come from different ERCs—thus, we can gain a geographically diverse understanding of the situation awareness of ERC personnel. One limitation is that the Swedish system is very much a diverse system when it comes to the roles and responsibilities and even education of the emergency response operators and emergency organisations' dispatchers. This may lead to a fragmented understanding of personnel's perspectives—perhaps, a greater number of interviews would have been necessary to cover more aspects of the study. The interviews were also performed via Zoom and Teams due to COVID-19, which can influence the quality of the interviews. The participants themselves could choose if they wanted to have a video call or if they did not want their picture showing, in order for them to feel as comfortable as possible. However, all in all, the participants gave a varied understanding of how it is to work during chemical incidents and the interviews were rich and in general about 1 hour in length, which

resulted in a lot of material to analyse. No limit was set for the period of time that had passed since the incident occurred as it is the content of the stories that responds to the purpose of the study.

### Practical implications and recommendations for future research

This study explores the situation awareness of the ERC personnel and identified challenges are discussed. In order to further increase the situation awareness of ERC personnel, a practical implication of this study could be to further develop emergency dispatch index interview guides that covers this type of incidents, so that it is more likely that the ERC personnel ask the most efficient questions. This could be an area where more research and development are needed.

Further practical implications could be to add a chemical incident database to the chemical substance database. In this database, all incidents could be documented together with their responses and lessons learnt and continually discussed in reflection seminars for ERC and emergency organisation personnel. Further research is needed of how such critical reflections could be implemented in practice and further improved during, for example, joint training and full-scale exercises. Such a database could theoretically also be a foundation to calculate resource allocation to such an incident.

### CONCLUSION

This study explores how ERC personnel attain accurate situation awareness of what is happening during potentially dangerous chemical incidents in order to be able to notify, inform and dispatch the correct resources. During chemical incidents there are several dichotomies. One is between the ERC personnel wanting as much information as possible for everyone's safety, also for emergency organisations, and their responsibility for the safety of the caller. This is not always a straightforward task, especially since chemical incidents can be considered vague. Another dichotomy is between using emergency dispatch index interview guides vs the ERC personnel trusting their gut feeling. Even if it would be possible to cover all probable chemical incidents in the emergency dispatch index interview guide it would still be necessary for the ERC personnel to trust their gut in the first place to even be able to identify the chemical incident. Thus, more research is needed to further explore these dichotomies.

**Acknowledgements** The authors would like to thank all participants for their contribution to the research. The authors would also like to thank professor Britt-Inger Saveman for helpful suggestions during the conception of the study and interview guide and recruitment of participants. The authors would also like to thank Raimo Lindgren for help with performing four of the interviews. We would also like to thank Johanna Björnstig for writing the transcriptions of the interviews.

**Contributors** SK and LG conceived and designed the study. SK interviewed all but four of the participants, which was performed by a research assistant. SK performed the first analysis and wrote the first draft of the manuscript both which was concurrently discussed with LG and revised. The final manuscript is approved by both SK and LG. LG is the guarantor.

**Funding** The research was funded by the Swedish National Board of Health and Welfare, grant number N/A.

**Competing interests** None declared.

**Patient and public involvement** Patients and/or the public were not involved in the design, or conduct, or reporting, or dissemination plans of this research.

**Patient consent for publication** Not applicable.

**Ethics approval** The study's objectives and methods are not covered by the Swedish Act concerning the Ethical Review of Research Involving Humans. Thus, approval from the Swedish Ethical Review Authority for this study has not been sought. Participants gave informed consent to participate in the study before taking part.

**Provenance and peer review** Not commissioned; externally peer reviewed.

**Data availability statement** All data relevant to the study are included in the article or uploaded as supplementary information.

**ORCID iDs**
Sofia Karlsson http://orcid.org/0000-0002-8665-9302
Lina Gyllencreutz http://orcid.org/0000-0002-1848-060X

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
