## [Reviewer comments · BMJ Open]

ARTICLE DETAILS

TITLE (PROVISIONAL)	Situation awareness of Emergency Response Centre personnel during chemical incidents – an interview study in a Swedish context
AUTHORS	Karlsson, Sofia; Gyllencreutz, Lina

VERSION 1 – REVIEW

REVIEWER	Mackway-Jones, Kevin Manchester Royal Infirmary, Emergency Department
REVIEW RETURNED	14-Feb-2023

GENERAL COMMENTS	This is an area which is of interest to specialist Staff and their managers. The study uses a very structured interview which is based around the recollection of the dispatcher of a particular type of incident. This incident is selected by the dispatcher rather than by the researcher, and there is no independent check of the veracity of the dispatchers recollections of the chosen incident. Further was no limit set for the period of time that had passed since the incident occurred The analysis is thorough The results of the analysis are presented in a clear and well written fashion The conclusions about the issued affecting situational awareness are informed by the interviews but appear to be more the opinions of the authors rather than the “results” of the study.
--

REVIEWER	Kirby, Kim South Western Ambulance Service NHS Foundation Trust, Research
REVIEW RETURNED	24-Feb-2023

GENERAL COMMENTS	General – Thank you for the opportunity to review this paper. You have investigated the situation awareness of ERC personnel when dealing with calls for a chemical incident. You used qualitative content analysis as your methods. I think this is an important and interesting topic and that your work is relevant to the readership of BMJOpen. I have attached some comments to assist in a minor revision of the paper. Abstract Emergency Response Centre (ERC) and then ERC throughout.
---

	Strengths and Limitations – make more succinct. Introduction Page 2 line 55 change to - Incidents can result from human error or poor maintenance of equipment used for manufacturing and storage. Page 4 11-15 – However, most chemical incidents are small or medium-sized and involve few killed or injured persons (1, 3). This is exemplified by a (can you name these incidents) or say recent incidents , rather than ‘a couple’. In the ‘setting’ section it would be good to have a better understanding of the Swedish system. Is it CBD, AMPDS, scripted dispatch. How is it organised? Are there call-takers and dispatchers, or just dispatchers? How many people does an ERC serve? Is an ERC operator a call-taker and is that different to a dispatcher? Use ERC throughout once defined. Participants – can you add table of participant demographics, roles, dual roles, experience. Patient and Public Involvement is duplicated in at the end of the paper. I think this section can be deleted. Data collection – who drafted the interview guide. There is a lot of information in this section which is then duplicated in the supplementary file as the interview guide is also here. I am not sure the manuscript needs this level of question detail here? I think this section could be more succinct. Data analysis – how did you manage the data – Excel, NVivo? Who transcribed the data? Ethical considerations is also at the end of the paper and is duplicated. Could this section be removed? Results When were the interviews completed and who participated? The detail about participants demographic etc should be in results. Page 8 – first paragraph of results – I think this could be shortened and picked up in the discussion. There is quite a lot in the literature about ‘intuition’ and what that is based on that could be referenced in the discussion. Also, there is a lot of literature on hysterical callers etc. Line 28 – correct in 95% of cases – says who, every participant. How would you know this is correct. Discussion Page 17 lines 12-15 – what are ‘personnel interview guides’? Line 46-48 – do you mean professional personnel, or emergency service personnel? When you say ‘other callers’ do you mean calls from bystanders? What do you mean by ‘third party caller’ -because I take it as someone not at the scene. P18, line 10 – change to - The subcategory, ‘The importance of ... Page 18 – lines 51-60 – Avoid using ‘another study’ use the author name perhaps? Strengths and limitations – I don’t think this needs to be repeated from your previous bullet points Practical Implications – change to ‘Practical Implications and recommendations for future research’ – The first sentence isn’t needed. Please make this section more succinct.
--	--

VERSION 1 – AUTHOR RESPONSE

Reviewer: 1 Prof. Kevin Mackway-Jones, Manchester Royal Infirmary Comments to the Author	Answers
The study uses a very structured interview which is based around the recollection of the dispatcher of a particular type of incident. This incident is selected by the dispatcher rather than by the researcher, and there is no independent check of the veracity of the dispatchers recollections of the chosen incident. Further was no limit set for the period of time that had passed since the incident occurred	We have pointed out that this is a qualitative research design and the participants' own experiences is in focus. One sentence is added in the method. This qualitative research is based on the narratives of the participants.
The analysis is thorough	Thank you
The results of the analysis are presented in a clear and well written fashion	Thank you
The conclusions about the issued affecting situational awareness are informed by the interviews but appear to be more the opinions of the authors rather than the "results" of the study.	The conclusion is rewritten to be more grounded in the results of the study.

Reviewer: 2

Ms. Kim Kirby, South Western Ambulance Service NHS Foundation Trust

Comments to the Author:

Thank you for the opportunity to review this paper. I have attached comments.

Emergency Response Centre (ERC) and then ERC throughout.	Thank you. Changed to the abbreviation throughout the manuscript.
Strengths and Limitations – make more succinct.	We have revised the strengths and limitations (bullet points) and the focus is more clearly related to the methods.
Introduction	
Page 2 line 55 change to - Incidents can result from human error or poor maintenance of equipment used for manufacturing and storage.	Thank you. Changed.
Page 4-15	

However, most chemical incidents are small or medium-sized and involve few killed or injured persons (1, 3). This is exemplified by a (can you name these incidents) or say recent incidents , rather than 'a couple'.	Changed to: exemplified by a fire in a Swedish battery factory and an ammonia spill that injured about 10 persons respectively (5, 6).
In the 'setting' section it would be good to have a better understanding of the Swedish system. Is it CBD, AMPDS, scripted dispatch. How is it organised? Are there call-takers and dispatchers, or just dispatchers? How many people does an ERC serve? Is an ERC operator a call-taker and is that different to a dispatcher?	We have described the ERC system in Sweden in more detail.
Use ERC throughout once defined.	We have changed to ERC throughout the manuscript.
Participants – can you add table of participant demographics, roles, dual roles, experience.	We see your point. A table might not be the best solution. In order to clarify who the participants were, we have added information to the content.
Patient and Public Involvement is duplicated in at the end of the paper. I think this section can be deleted.	Thank you. Patient and public involvement is deleted in the method section.
Data collection – who drafted the interview guide. There is a lot of information in this section which is then duplicated in the supplementary file as the interview guide is also here. I am not sure the manuscript needs this level of question detail here? I think this section could be more succinct.	We have removed duplications. Added who drafted the interview guide and made the section more concise.
Data analysis – how did you manage the data – Excel, NVivo? Who transcribed the data?	Microsoft Excel 2019 is added.

Ethical considerations is also at the end of the paper and is duplicated. Could this section be removed?	Removed duplications.
Result	
When were the interviews completed and who participated? The detail about participants demographic etc should be in results.	The year of when the interviews were completed is added. Thank you for the suggestion to move the detail about participants demographic to the result. However, we choose to keep the demographic in the method section. We hope that is fine with you.
Page 8 – first paragraph of results – I think this could be shortened and picked up in the discussion. There is quite a lot in the literature about ‘intuition’ and what that is based on that could be referenced in the discussion. Also, there is a lot of literature on hysterical callers etc. Line 28 – correct in 95% of cases – says who, every participant. How would you know this is correct.	We have shortened the first paragraph in the results section. The content of gut feeling is discussed in the first part of the discussion.
Discussion	
Page 17 lines 12-15 – what are ‘personnel interview guides’?	Changed to emergency dispatch index interview guide
Line 46-48 – do you mean professional personnel, or emergency service personnel? When you say ‘other callers’ do you mean calls from bystanders?	The section is rewritten to be clearer.
What do you mean by ‘third party caller’ - because I take it as someone not at the scene.	Third party caller is removed. The section rewritten to be clearer.

P18, line 10 – change to - The subcategory, 'The importance of ...	We have rewritten the sentence.
Page 18 – lines 51-60 – Avoid using 'another study' use the author name perhaps?	We have changed other studies and instead used the name of the author.
Strengths and limitations	
I don't think this needs to be repeated from your previous bullet points	We deepen the reasoning of the bullet points as they only are short statements
Practical Implications	
change to 'Practical Implications and recommendations for future research' – The first sentence isn't needed. Please make this section more succinct.	Changed as suggested. The first sentence is removed and the section is now more truthful.